# Exploring the Causes of the Cambrian Explosion Based on the Evolution Mechanism of Genome Sequences

**DOI:** 10.3390/biology14070783

**Published:** 2025-06-27

**Authors:** Xiaolong Li, Hong Li, Zhenhua Yang, Qiang Zhang, Liaofu Luo

**Affiliations:** 1College of Science, Inner Mongolia Agriculture University, Hohhot 010018, China; ndlixiaolong@126.com (X.L.); zhangqiang829@163.com (Q.Z.); 2Inner Mongolia Autonomous Region Key Laboratory of Biophysics and Bioinformatics, School of Physical Science and Technology, Inner Mongolia University, Hohhot 010021, China; lolfcm@imu.edu.cn; 3School of Economics and Management, Inner Mongolia University of Science and Technology, Baotou 014010, China; zhyang_83@126.com

**Keywords:** evolution mechanism of genome sequences, independent selection modes, Cambrian explosion, phase transition process, origin time of species branches

## Abstract

**Simple Summary:**

This study investigated a possible cause of the Cambrian explosion—a rapid increase in biodiversity around 540–560 million years ago—by analyzing evolution modes of genome sequences in modern species. Specifically, we examined two evolutionary modes: CG- and TA-independent selection, both of which play critical roles in shaping genome sequence changes across different evolutionary levels of species. Our findings suggest that the transition from TA-independent selection dominance to CG-independent selection dominance may have acted as a catalyst for this ancient biological event. By estimating evolution timelines based on this transition, we reconstructed the origins of major species groups, which are consistent with existing paleontological evidence. This methodology offers a novel framework for studying ancient evolution events through the lens of genome sequence data from modern species.

**Abstract:**

The cause of the Cambrian explosion is one of the centuries-old puzzles. For centuries, scholars from numerous disciplines have proposed various theories based on evidence such as paleontological fossils and changes in geology and climate to try to reveal the cause of the Cambrian explosion, but no satisfactory conclusion has been reached. We explored a possible cause of the Cambrian explosion based on the evolution mechanism of genome sequences of existing species. Previous studies have found that the CG- and TA-independent selection intensities and the mutual inhibition relationship between them determine the evolution state of genome sequences. Based on the evolution mechanism of genome sequences, we analyzed the distribution of CG- and TA-independent selection intensities in animals and plants. We believed that the phase transition process from the evolution mode dominated by TA-independent selection to that dominated by CG-independent selection is an important cause of the Cambrian explosion. Consequently, we deduced the evolution time corresponding to the evolution state of genome sequences and gave the origin time of species branches. The results are largely consistent with existing paleontological evidence for animal branches and some plant branches, which verifies the rationality of our conjecture, though differences for certain plant groups require further investigation. Our study provides a novel way to reveal the cause of the Cambrian explosion and the origin time of species branches through existing genome sequences.

## 1. Introduction

It has been 4600 million years since the formation of the Earth, and life began to appear only a few hundred million years after the formation of the Earth. The evolution of life has experienced three major revolutionary events. First, of course, was the birth of life, estimated to have occurred 4000 million years ago (Ma). The second is the leap from prokaryotes to simple eukaryotes, which occurred around 2000 Ma. The third is the evolution from protozoa to metazoa, which occurred around 560–520 Ma, known as the Cambrian explosion. During tens of millions of years (less than 1% of the Earth’s history) in the period [1], a wide variety of multicellular organisms “suddenly” appeared on Earth. Paleontologists refer to this “instantaneous” increase in species diversity as the Cambrian explosion. Many scholars in paleontology, biology, geology, paleoclimatology and other disciplines have proposed various theories to try to reveal the mystery of the Cambrian explosion, but this problem has not been satisfactorily resolved, and it has been classified as one of the scientific problems by the international academic community.

The conclusions of the study on the mystery of the Cambrian explosion are mainly reflected in both the external environmental driver and the internal evolutionary driver of the organisms [2]. It is believed that drastic changes in the living environment are essential conditions as the external driver of life evolution [3,4,5,6]. As early as 1964, American physicists Berkner and Marshall suggested that the Cambrian explosion event was caused by an increase in the oxygen content of the environment. Photosynthesis by algae caused a rapid rise in the oxygen content of the atmosphere and oceans, which favored the survival of larger and more complex animals. At the same time, the continuous improvement of the ozone layer helped protect life in shallow oceans from UV damage [3]. In 2004, American geologist Brennan et al. argued that a large influx of calcium in the ocean led to the Cambrian explosion. They found that calcium levels were at least three times higher during the early Cambrian period than before the Cambrian period. They believe the calcium surge allowed early life to evolve shells and skeletons to cope with the potential toxicity of higher calcium levels in seawater, leading to the Cambrian animal boom [4]. In 2018, Wang et al. showed that ocean redox state during the Ediacaran–Cambrian transition about 551–515 Ma showed stage-specific fluctuations in timing that highly coincided with stage-specific radiation and extinction events of early metazoan. This research result provides a new perspective on the mystery of the Cambrian explosion [5]. In 2019, a study by He et al. used quantitative models of carbon and sulfur isotopes to argue for the first time that the Cambrian explosion process in animals was controlled by changes in atmospheric and oceanic oxygen levels [6]. It was shown that during the peak of the Cambrian explosion, which occurred about 524–514 Ma in the early Cambrian period, the number and amplitude of changes in oxygen levels of the atmosphere and shallow ocean were highly coincident with the number and amplitude of changes in faunal fossil diversity. During the about 2 million years after 514 Ma, there was oceanic anoxia, which coincided with the Botoman–Toyonian animal extinction events.

Although external environmental factors are important in driving the evolution of life, it is life itself that does so. Life’s own evolution mechanisms played a key role as an internal driver in the Cambrian explosion [7,8,9,10]. In 1973, American ecologist Stanley proposed the biological harvesting hypothesis [7]. He believed that during the Precambrian period, the ocean was dominated by prokaryotic cyanobacteria and no predation relationships existed, resulting in slow community evolution. During the Cambrian period, protozoan predators that consumed prokaryotic cyanobacteria emerged, and producers diversified in order to survive, which in turn led to the evolution of predators, thus enriching the biodiversity of the entire ecosystem and eventually leading to the Cambrian explosion. Some researchers have suggested that sexual reproduction may have been an important factor in the Cambrian explosion. Precambrian fossil data indicate that multicellular sexual reproduction (red algae) occurred as early as the Ectasian period (1400–1200 Ma) [8]. However, the earliest multicellular animal with sexual reproduction discovered so far is a slender rope-like organism called *Funisia dorothea* that lived 570 Ma [9]; this was just before the Cambrian period. The phenomenon of sexual reproduction could provide more possibilities for genetic variability and further increase biodiversity, resulting in the Cambrian explosion. In 2018, Hammarlund and his colleagues from Sweden proposed a new hypothesis for the Cambrian explosion [10]. They suggest that multicellular life can manipulate a cellular protein that allows cells to perceive themselves as low oxygen levels in a high oxygen environment, thereby maintaining stem cell properties that allow life to cope with fluctuating oxygen concentrations. It is this mechanism that prevents tissue-specific stem cells from maturing too early due to high oxygen levels, thus facilitating the rapid growth of Cambrian life. Additionally, phylogenomic studies have utilized large-scale genomic data to reconstruct deep evolutionary relationships, providing insights into the timing and mechanisms of early animal diversity [11,12]. Molecular clock analyses calibrated with fossil records further refined estimates of divergence times for major metazoan lineages; it indicated that genetic innovation may have preceded morphological diversity during the Cambrian explosion [13].

We know that most of the information about life is contained in genome sequences. At each stage of life evolution, whether it is external driver of the environment or internal driver of the organism itself, it must be reflected in the evolutional process of genome sequences. The revolutionary event of the Cambrian explosion must be reflected in the revolutionary changes of evolution modes in the genome sequences. Therefore, exploring the changes of evolution modes of genome sequences in the Cambrian period is the most fundamental way to reveal the Cambrian explosion. However, we are unable to obtain the genome sequences of organisms that existed during the Cambrian period. It would be a theoretical breakthrough if we could infer the evolutionary laws of species during the Cambrian period through existing genome sequences. To address this, we assumed that the evolution mechanisms observed in modern genome sequences, specifically the CG- and TA-independent selection modes, have persisted since the Cambrian period, though we recognize that factors such as genetic drift, lineage-specific adaptations and incomplete sampling may introduce uncertainties in these inferences. Our previous work investigated the spectral distribution rule of 8-mer subsets of more than 1000 genome sequences ranging from prokaryotes to primates and found that there were CG-independent selection mode and TA-independent selection mode in genome sequences [14]. Therefore, we proposed an evolution mechanism of genome sequences. We will analyze the distribution patterns of CG- and TA-independent selection intensities of species genome sequences. Drawing inspiration from stellar evolution models in astronomy, we use a meta-organism approach to infer historical evolutionary processes. We acknowledge that this analogy is heuristic, but it requires careful validation due to differences between biological and stellar systems. Based on the evolution mechanism of genome sequences and the idea of deducing stellar evolution in astronomy, we proposed a conjecture of the Cambrian explosion and deduced the origin time of species branches.

## 2. Materials and Methods

### 2.1. Dataset

The whole genome sequences and annotated information of all species involved in this study were obtained from NCBI (https://www.ncbi.nlm.nih.gov/ (accessed on 24 April 2023)) and CNGBdb (https://db.cngb.org/ (accessed on 24 April 2023)). The selected species number is 1087. Animals are divided into 8 branches and plants are divided into 6 branches; see Table 1. The detailed information is provided in Additional file: Appendix A.

### 2.2. 8-mer Spectrum

For a given genome sequence, with 8 bp as the window and 1 bp as the step size, the frequency of each 8-mer in the sequence is obtained statistically. If the number of 8-mers that occur *i* times is *N_i_*, the 8-mer relative motif number (*RMN*) is defined as(1)$$RMN=\frac{N_{i}}{4^{8}},$$

After obtaining the frequency of each 8-mer in a genome sequence, the 8-mers were classified by the XY dinucleotide classification method according to the composition characteristics of the 8-mer fragments. The 8-mers without XY (X, Y = A, T, C, G) dinucleotide were assigned to the XY0 subset, those containing one XY dinucleotide were assigned to the XY1 subset, and those containing two or more XY dinucleotides were assigned to the XY2 subset. Theoretically, there are 4^8^ = 65,536 8-mers. When X ≠ Y, the number of 8-mers in XY0 subsets is 40,545, in XY1 subsets is 21,468, and in XY2 subsets is 3523. When X = Y, the number of 8-mers in the three kinds of subsets is 44,631, 14,931 and 5974, respectively. With the occurrence frequency of 8-mers as the horizontal axis and the *RMN* as the vertical axis, the spectral distributions of total 8-mers and 8-mers of each XY*i* subset are obtained, respectively. For each species genome sequence, we can obtain 64 8-mer spectra for subsets and one 8-mer spectrum for total 8-mers.

### 2.3. Separability of Subset Spectrum

The average and standard deviation of the spectral distribution is used to represent its distribution characteristics. In order to eliminate the influence of different genome sizes and show the relative position difference of 8-mer spectra in different subsets, the separability values (*δ*_XY_*_i_*) are defined.(2)$$\delta_{XYi}=\frac{\bar{x}}{{\bar{x}}_{XYi}},$$
where $\bar{x}$ is the average frequency of total 8-mers in a genome sequence and is called the random center. ${\bar{x}}_{XYi}$ is the average frequency of XY*i* 8-mers. *δ*_XY_*_i_* represents the separability for the distribution position of XY*i* 8-mer spectra relative to the random center. If *δ*_XY_*_i_* > 1, it indicates that XY*i* 8-mer spectra are located at the low-frequency end of the random center. If *δ*_XY_*_i_* = 1, it indicates that the location of XY*i* 8-mer spectra does not separate from the random center. If *δ*_XY_*_i_* < 1, it indicates that XY*i* 8-mer spectra are located at the high-frequency end of the random center. The characteristic parameter of the separability can be used to compare not only the relative separation degree of 8-mer spectra in different subsets within a genome sequence but also among different genome sequences.

### 2.4. Conservatism of Subset Spectrum

In order to eliminate the influence of different genome sizes and show the relative discrete degree of the 8-mer spectral distribution of each subset, the conservatism values (*ρ*_XY_*_i_*) of subset 8-mer spectra are defined.(3)$$\rho_{XYi}=\frac{SD}{{SD}_{XYi}},$$
where *SD* is the standard deviation of spectral distribution for total 8-mers, and *SD*_XY_*_i_* is the standard deviation of XY*i* 8-mer spectra. *ρ*_XY_*_i_* represents the relative value of standard deviation of XY*i* 8-mer spectra. If *ρ*_XY_*_i_* > 1, it indicates that the distribution of XY*i* 8-mer spectra is more conservative than that of total 8-mer spectrum; that is to say, the distribution of 8-mer frequencies in this subset is more concentrated. The characteristic parameter of the conservatism can be used to compare not only the relative conservation difference of 8-mer spectral distribution in different subsets within a genome sequence but also among genome sequences.

### 2.5. Evolution Mechanism of Genome Sequences

Previous and subsequent works studied the spectral distribution for various 8-mer subsets in more than 1000 genome sequences ranging from prokaryotes to primates [14]. It is found that the evolution differences of genome sequences are mainly reflected in the separability and conservatism of the CG*i* and TA*i* 8-mer spectra. The separability and conservatism of the other 14 XY*i* 8-mer spectra are not related obviously to the evolution differences. It indicated that there are two evolution modes in genome sequences. Finally, the evolution mechanism of genome sequences was proposed. It is that the CG- and TA-independent selection intensities and the mutual inhibition relationship between them determine the evolution state of genome sequences. Considering the positive correlation between the separability and conservatism, the CG- or TA-independent selection intensity is defined by the separability of the CG1 8-mer spectrum or the TA1 8-mer spectrum. It is found that the CG-independent selection intensity is positively correlated and the TA-independent selection intensity is negatively correlated with the evolutionary levels of animals and plants.

Considering that the CG1 and CG2 8-mer spectra have the same evolutionary property, in this study, the two subsets CG2 and CG1 are merged into one subset called the CG1 subset. We made the same simplification for the TA1 and TA2 subsets called the TA1 subset. We used the CG-independent selection intensity, which comes from the new CG1 subset, or the TA-independent selection intensity, which comes from the new TA1 subset, to characterize the evolution state of genome sequences.

### 2.6. Relationship Between the Evolution State (δ_CG1_) and the Evolution Time

The CG-independent selection intensity was used to characterize the evolution state of genome sequences. It is known that the evolution state *δ*_CG1_ has a nonlinear positive correlation with evolution time. We acknowledge that the true relationship between evolution rate *v* and the evolution state *δ*_CG1_ may be nonlinear, as shown in the data. However, for ease of analysis, we adopted the linearity assumption as the first-order approximation. This simplification allowed us to establish a tractable model that provides a basis for subsequent studies. Nevertheless, we recognize that this assumption may not capture all the complexity of the evolutionary process. Future studies could explore nonlinear models or more complex statistical methods to describe evolutionary dynamics more accurately. As the most basic consideration, we assumed that there is a linear positive correlation between the evolution rate *v* and the evolution state *δ*_CG1_; that is:(4)$$v=k\delta_{CG1},$$
here the proportional coefficient *k* is the constant, and the evolution rate *v* is defined as:(5)$$v=\frac{d\delta_{CG1}}{dt},$$
it follows that:(6)$$\frac{d\delta_{CG1}}{dt}=k\delta_{CG1},$$
integrating Equation (6):(7)$$\int_{\delta_{0}}^{\delta_{CG1}}\frac{1}{\delta_{CG1}}d\delta_{CG1}=\int_{t_{0}}^{t}kdt,$$(8)$$\delta_{CG1}=\delta_{0}e^{k(t-t_{0})},$$

Equation (8) gives a nonlinear relationship between the evolution state and the evolution time.

According to our analysis, the time corresponding to *δ*_CG1_ = *δ*_0_ = 1 (phase transition point) is denoted as the starting time of the Cambrian explosion *t* = *t*_0_ = −*τ*. *τ* = 560–520 million years, which is the time since the Cambrian explosion. The time corresponding to *δ*_CG1_ = *δ*_CG1,max_ is denoted as the present, it is *t* = 0. By putting them into Equation (8), the proportional coefficient *k* can be obtained:(9)$$k=\frac{1}{\tau }ln\frac{\delta_{CG1,max}}{\delta_{0}},$$
according to Equation (8), the evolution time corresponding to the evolution state of each species is obtained:(10)$$t=-\tau +\frac{1}{k}ln\frac{\delta_{CG1}}{\delta_{0}},$$

### 2.7. Relationship Between the Evolution State (δ_TA1_) and the Evolution Time

For green algae, the evolution mode of genome sequences is dominated by TA-independent selection, and TA-independent selection intensity can also be used to characterize the evolution state. It is known that the TA-independent selection intensity *δ*_TA1_ has a nonlinear negative correlation with evolution time. Similarly, as the most basic consideration, we assumed that there is a negative linear relationship between the evolution rate *v*′ and the evolution state *δ*_TA1_, and the proportional coefficient is the same as the evolution time characterized by the CG-independent selection intensity; that is:(11)$$k^{\prime }=k=\frac{1}{\tau }ln\frac{\delta_{CG1,max}}{\delta_{0}},$$
so(12)$$v^{\prime }=k\delta_{TA1},$$
this assumption is also intended to simplify the analysis, but we recognize that the true relationship may be more complex. Future studies may consider nonlinear models or other methods to describe the evolutionary dynamics of green algae more accurately. Here, *δ*_TA1_ decreases with time *t*, so the evolution rate *v*′ should be as follows:(13)$$v^{\prime }=-\frac{d\delta_{TA1}}{dt},$$
it follows that:(14)$$-\frac{d\delta_{TA1}}{dt}=k\delta_{TA1},$$
integrating Equation (14):(15)$$\int_{\delta_{0}^{\prime }}^{\delta_{TA1}}-\frac{1}{\delta_{TA1}}d\delta_{TA1}=\int_{t_{0}^{\prime }}^{t^{\prime }}kdt,$$(16)$$\delta_{TA1}=\delta_{0}^{\prime }e^{-k(t^{\prime }-t_{0}^{\prime })},$$

The phase transition point corresponding to the Cambrian explosion is still selected as *δ*_CG1_ = *δ*_0_ = 1. It is found that the corresponding TA-independent selection intensity value is *δ*_TA1_ = 2. The time corresponding to *δ*_TA1_ = *δ*_0_′ = 2 is denoted as the starting time of the Cambrian explosion *t*′ = *t*_0_′ = −*τ*. According to Equation (16), the evolution time corresponding to the evolution state of each species is obtained:(17)$$t^{\prime }=-\tau -\frac{1}{k}ln\frac{\delta_{TA1}}{\delta_{0}^{\prime }}$$

## 3. Results

### 3.1. Characterization of the Evolution State of Genome Sequences

Previous work has shown that there are CG- and TA-independent selection modes in genome sequences [14]. These two kinds of evolution modes can be characterized by the separability values of the CG1 and TA1 8-mer spectra (Section 2), which are called the CG-independent selection intensity (*δ*_CG1_) and the TA-independent selection intensity (*δ*_TA1_). There is a mutual inhibition relationship between the two independent selection intensities. With the increase of species evolutionary levels, the CG-independent selection intensity increases, but the TA-independent selection intensity decreases in animals and plants. Therefore, an evolution mechanism of genome sequences was proposed; that is, the CG- and TA-independent selection intensities and the mutual inhibition relationship between them determine the evolution state of genome sequences. It is believed that the mutual inhibition mechanism is the key reason driving the evolution of genome sequences. To evaluate the robustness of the independent selection intensities, we conducted a resampling analysis. Specifically, we performed multiple sampling with replacement on the animal dataset, generating multiple subsamples. For each pair of subsamples, we calculated the CG-independent selection intensities and performed linear regression analysis. The Pearson correlation coefficient (R) ranged from 0.898 to 0.999, and *p*-values ranged from 1.94 × 10^−93^ to 1.05 × 10^−26^. These results indicated that there is a statistically significant correlation between independent selection intensities across different subsamples. This means that the CG-independent selection intensity parameter is highly robust in different data subsets.

Here, animals are divided into eight branches: invertebrates, fishes, amphibians, reptiles, birds, other mammals, rodents and primates, and plants are divided into six branches: green algae, mosses, ferns, monocotyledons, dicotyledons and modern gymnosperms. The distributions of CG- and TA-independent selection intensities for their genome sequences are given, respectively (Figure 1). The detailed information is listed in Additional file: Appendix A.

In animal genome sequences, it can be seen that the CG-independent selection intensity is positively correlated with the evolutionary levels of animals (Figure 1B). In invertebrates, fishes, amphibians and reptiles, there is a mutual inhibition relationship between the CG- and TA-independent selection intensities. Starting from other mammals, with the increase in species evolutionary levels, the TA-independent selection mode gradually disappears (*δ*_TA1_ ≈ 1). In plant genome sequences, the CG-independent selection intensity is positively correlated with the evolutionary levels of plants (Figure 1D), and the mutual inhibition relationship exists in all plant branches.

We found that the distribution pattern of a few lower animals is *δ*_TA1_ > 1 and *δ*_CG1_ < 1, and that of other animals is *δ*_CG1_ > 1 and *δ*_TA1_ < 1. In plants, the distribution pattern of some green algae is *δ*_TA1_ > 1 and *δ*_CG1_ < 1, that of some green algae, some mosses, some ferns and some monocotyledons is *δ*_CG1_ > 1 and *δ*_TA1_ > 1, and that of other plants is *δ*_CG1_ > 1 and *δ*_TA1_ < 1. Overall, the CG- and TA-independent selection intensities are different for different species. Here, we use the CG- or TA-independent selection intensity to characterize the evolution state of species genome sequences.

### 3.2. Conjecture on the Cause of the Cambrian Explosion

The Cambrian explosion is a revolutionary event in the history of biological evolution, which has the following features in its macroscopic manifestations: (1) the “sudden” prosperity of eukaryotic species; (2) the “sudden” increase in complexity of eukaryotic evolution. From the perspective of genome sequence evolution, we argue that the “sudden” increase in biodiversity and structural complexity of organisms must be reflected in revolutionary changes in the way of genome sequence evolution.

Darwin’s theory stated that higher organisms evolved from lower organisms. According to the distribution patterns of CG- and TA-independent selection intensities along with the evolutionary levels of animals and plants, we found that if we do not distinguish between species, all animals or all plants can be regarded as one kind of organism, which we call the meta-organism. Then, the arrangement of animals or plants according to their evolutionary levels can be regarded as the evolution timeline of the meta-animal or meta-plant (Figure 1A,C). Then, the change in the CG-independent selection intensity from small to large or the TA-independent selection intensity from large to small corresponds to the change in the evolution time from past to present for the meta-organism. Thus, the evolution time is related to the evolution state for the meta-organism. This approach is inspired by the method used in astronomy to deduce stellar evolution. By observing the “present” image of stars, astronomers analyze the feature differences of stars according to physics theory and deduce their evolutionary history. Similarly, genome sequences of all existing species unfold before our eyes, like the observed “present” image of stars. By analyzing the differences in the evolution state among different species, we can deduce the evolutionary process of the meta-organism based on the evolution mechanism of genome sequences. This meta-organism model was inspired by stellar evolution theories in astronomy, which consider the collective evolution states of modern species as historical evolutionary trajectories. However, unlike stellar systems that strictly follow physical laws, the evolution mechanisms of biological systems involve complex factors such as genetic drift, selection pressure and extinction events. This essential difference makes the direct mapping of astrophysical models in the biological field theoretically limited. We address these limitations by basing our inferences on a model that relies on CG- and TA-independent selection intensities across diverse groups.

When the arrangement of animals or plants by their evolutionary levels is regarded as the evolution timeline of the meta-organism, we found that the distribution curves of the CG-independent selection intensity and the TA-independent selection intensity intersected at a certain time in the past (Figure 1). The species before and within this intersection region are some lower invertebrates or green algae, while the species after this intersection region are higher eukaryotes. This indicates that in the evolutionary process from lower organisms to higher organisms, there is a transition process from the evolution mode dominated by TA-independent selection to the evolution mode dominated by CG-independent selection. We call this phenomenon the phase transition process of evolution modes. According to the distributions in Figure 1, we can see that genome sequences adopted the evolution mode dominated by TA-independent selection in the early stage of life evolution, while the CG-independent selection mode was inhibited. As life evolved further, the CG-independent selection intensity gradually increased. Eventually, genome sequences adopted the evolution mode dominated by CG-independent selection, while the TA-independent selection was inhibited or even disappeared.

For the Cambrian explosion, this revolutionary event must be reflected in revolutionary changes in the evolution modes of genome sequences. The phase transition process between the TA- and CG-independent selection modes is a revolutionary event in the evolutionary process of genome sequences. We assumed that the evolution modes of CG- and TA-independent selection have persisted since the Cambrian period. This allows us to infer the evolution states of ancient organisms from modern genome sequences. However, we acknowledge potential limitations, including the possibility that genetic drift, lineage-specific adaptations, or incomplete sampling of modern genomes may obscure ancient evolutionary information. We believe that the evolution rule is immutable from past to present in genome sequences. Although it is impossible to obtain the genome sequences of species in the Cambrian period, we consider that the phase transition phenomenon exhibited by existing species is the main cause of the Cambrian explosion. This phenomenon originated in the Cambrian period and continues to the present. The evolution of organisms from simple to complex eukaryotes is determined by the phase transition phenomenon. Based on our conjecture, we will analyze the relationship between the evolution state and the evolution time of species and deduce the origin time of species branches.

### 3.3. Relationship Between the Evolution Time and the Evolution State

Considering that the CG-independent selection intensity changes more obviously across species, here we use it to characterize the evolution state of genome sequences. First, we need to determine the two evolution states of genome sequences corresponding to the starting time of the Cambrian explosion and the present time. Second, we need to determine the function that describes the relationship between the evolution time and the evolution state of genome sequences.

We found that regardless of animals or plants, there is such a rule for the transition of evolution mode in the phase transition region: starting from *δ*_TA1_ > 1 and *δ*_CG1_ < 1, going through the intermediate process of *δ*_TA1_ > 1 and *δ*_CG1_ > 1, and finally reaching the evolution state of *δ*_CG1_ > 1 and *δ*_TA1_ < 1 (Figure 1A,C). According to the definition of independent selection intensity (see “Section 2”), when *δ*_CG1_ ≠ 1 or *δ*_TA1_ ≠ 1, the distribution location of the 8-mer spectrum in CG1 or TA1 subsets is apart from the random center. Taking *δ*_CG1_ as an example, when *δ*_CG1_ > 1, the 8-mer spectrum of the CG1 subset is located at the low-frequency end of the random center. With *δ*_CG1_ increases, the distribution conservatism of the 8-mer spectrum of the CG1 subset also increases. When *δ*_CG1_ < 1, the 8-mer spectrum of the CG1 subset is located in the high-frequency end of the random center. With *δ*_CG1_ decreases, the distribution conservatism of the 8-mer spectrum of the CG1 subset also decreases [14]. The 8-mer spectral properties of TA1 subsets are the same as those of CG1 subsets. If the 8-mer spectral distribution has high conservatism and is far from the random center, these 8-mers must be functional motifs [15,16]. Our results showed that *δ*_CG1_ is obviously positively correlated with the evolutionary levels of genome sequences, and the TA-independent selection mode disappears basically (*δ*_TA1_ ≈ 1) in mammals (Figure 1B,D). This indicated that the CG-independent selection mode is a driving force in the evolution of genome sequences, while the TA-independent selection mode is the result of being inhibited. In conclusion, it is reasonable to choose *δ*_CG1_ = 1 as the critical point of the phase transition between two evolution modes of animal and plant genome sequences and to choose the time corresponding to *δ*_CG1_ = 1 as the starting time of the Cambrian explosion. This choice assumes that the CG- and TA-independent selection intensities observed in modern genomes reflect the evolutionary dynamics of the Cambrian period.

In principle, the selection of the evolution state of the meta-organism genome sequence corresponding to the present time should satisfy two conditions: one is that the evolutionary level of the selected species should be the highest, and the other is that the CG-independent selection intensity of the selected species should be the largest. In animals, it is generally assumed that the maximum value *δ*_CG1,max_ should appear in primates with the highest evolutionary level. However, as shown in Figure 1D, we can see high average values of CG-independent selection intensity in primates and rodents, but the highest value is found in other mammals. The maximum values of the CG-independent selection intensity in rodents and primates are obviously lower than in other mammals. This brings confusion to the choice of *δ*_CG1,max_. However, in Figure 1D, we found that the TA-independent selection mode is weak (*δ*_TA1_ ≈ 1) in other mammals, and the TA-independent selection mode disappears completely in rodents and primates. That is to say, the mutual inhibition relationship between the two evolution modes is weak or disappears in these genome sequences. The results showed that a new evolution mechanism arose in primates and rodents. Due to the disappearance of the mutual inhibition relation, they no longer obey the general evolution mechanism we proposed. Therefore, the changes in CG-independent selection intensity are no longer dependent on the mutual inhibition relationship.

Therefore, the following analysis does not consider these two branches. It is reasonable for us to select *δ*_CG1,max_ from other mammals. Finally, in animals, we obtained *δ*_CG1,max_ = 12.45, which comes from *Sarcophilus harrisii* and represents the highest level of the evolution state. We use the evolution state of *Sarcophilus harrisii* to represent the evolution state of the meta-animal. We record the time corresponding to the evolution state of the meta-animal as the present time. In plants, we obtained *δ*_CG1,max_ = 6.04, which comes from *Sequoia sempervirens* and represents the highest level of the evolution state. We use the evolution state of *Sequoia sempervirens* to represent the evolution state of the meta-plant. We record the time corresponding to the evolution state of the meta-plant as the present time.

In animals and plants, the arrangement of species was regarded as the evolution timeline of the meta-animal or meta-plant (Figure 1A,C). We found that there is a nonlinear positive correlation between the evolution time and the evolution state of the meta-animal or meta-plant. This means that as the CG-independent selection intensity increases, so does the evolution rate (slope of distribution) of species. Therefore, it is inappropriate to directly characterize the evolution time by the CG-independent selection intensity. In Figure 1B,D, if we pay attention to the overall trend of the distribution of the CG-independent selection intensity for each animal branch or plant branch, we found that there is a linear positive correlation between the evolution state and the evolution time. This is also reflected in the first half of the distribution in Figure 1A,C. If the evolution rate of the overall trend of the distribution is denoted as the *v*_0_ value, the evolution rate *v*_0_ characterizes the commonality of the evolution of genome sequences across each animal or plant branch. The evolution rates vary widely among species within each animal branch (Figure 1B). With some exceptions among invertebrates and fishes, the evolution rates of other species are obviously higher than *v*_0_. The evolution rates also differ within each plant branch (Figure 1D), such as mosses, ferns and modern gymnosperms; their evolution rates are obviously higher than *v*_0_. This difference represents the individuality of the evolution of genome sequences and is closely related to the differences in the living environment of species. In conclusion, the relationship between the evolution state and the evolution time is nonlinear. When exploring the evolution of genome sequences, the commonality and individuality of species evolution must be considered at the same time.

According to the above analysis, we established a functional relationship between the evolution state and the evolution time. As a basic consideration, we assumed that there is a linear positive correlation between evolution rate *v* and the evolution state, that is, *v* = *kδ*_CG1_, where the proportional coefficient *k* is a constant. The evolution rate is defined as *v* = *dδ*_CG1_/*dt*. If the starting time of the Cambrian explosion is defined as *t* = *t*_0_, the corresponding evolution state is defined as *δ*_CG1_ = *δ*_0_ = 1 at the phase transition point. The present time is defined as *t* = 0, the corresponding evolution state of animals is defined as *δ*_CG1,max_ = 12.45, and the corresponding evolution state of plants is defined as *δ*_CG1,max_ = 6.04. After deduction (Section 2), we calculated the proportional coefficient *k* of animals and plants, respectively. Finally, we obtained the function relationship between the evolution state *δ*_CG1_ and the evolution time *t*, see Equation (10).

Since the accepted time for the Cambrian explosion event is 560–520 Ma [1], we present two sets of time *t* values to represent the time range corresponding to the evolution state of each species. We calculated the evolution time corresponding to the evolution state of genome sequences in 649 animals and 350 plants. The results are shown in Figure 2A,B and Additional file: Appendix A. In Figure 2A,B, the upper curve corresponds to the starting time of 560 Ma, and the lower curve corresponds to the starting time of 520 Ma. For example, for *Lithobates catesbeianus*, the CG-independent selection intensity is *δ*_CG1_ = 2.93, and the corresponding time *t* is 321–298 Ma, indicating that the evolution state of this species is equivalent to that of the meta-animal at 321–298 Ma.

### 3.4. The Origin Time of Each Species Branch

Next, we analyze the origin time of each species branch according to the relationship between the evolution state and the evolution time. Within each species branch, the species with the smallest CG-independent selection intensity should appear earliest. The appearance time of this species can represent the origin time of this species branch, and the evolution state of its genome sequence represents the morphology of the ancestral genome sequence of this species branch. According to this idea, we gave the origin time of six animal branches and six plant branches (Figure 2). The origin time of each animal branch is shown in Table 2, and the origin time of each plant branch is shown in Table 3. The tables also provide representative species of species origin in each branch. It is known that these representative species belong to the ancient species within their corresponding branch. For example, among reptiles, *Sphenodon punctatus* is a lizard-like animal of the genus *Sphenodon*, which is the only representative of the *Rhyncocephalia* remaining from the early Triassic period and is known as a living fossil. Among other mammals, Ornithorhynchus anatinus belongs to monotremes or oviparous mammals, which represents a link in the evolution from reptiles to mammals and is considered to be the oldest mammal.

Among animals, we compared the origin time of the six animal branches with paleontological evidence and found that they are extremely consistent. For invertebrates, paleontological evidence showed that the Ediacaran biota emerged during the Precambrian period (575–542 Ma) [17,18]. The origin time we deduced is 579–537 Ma. For fishes, paleontologists have discovered an ancient jawed fish named *Bianchengichthys micros* that existed 423 Ma [19]. More recently, Zhu’s team has discovered a variety of early Silurian jawed fish fossils in the geological time of 439–436 Ma [20,21,22,23]. The origin time we obtained is 429–398 Ma, which is in the late Silurian and early Devonian. For amphibians, paleontologists believe that *Ichthyostega* is the earliest known amphibian [24,25]. Its fossils were discovered in strata from the late Devonian period, about 370–360 Ma. *Elpistostege watsoni* [26,27,28] and *Tiktaalik roseae* [29], which predate *Ichthyostega*, had their fossils found in late Devonian strata around 380 Ma. They have both fish and amphibian characteristics and are considered to be transitional species between fish and tetrapods. So paleontological evidence suggests that amphibians originated around 380–360 Ma. The origin time we gave is 391–363 Ma, which is in the Devonian period. For reptiles, scientists discovered fossil footprints dating back 318 Ma that belong to *Hylonomus lyelli* [30,31], a small lizard-like animal from the Carboniferous period. However, its bone fossils were discovered in strata of 314 Ma. Paleontological evidence suggests that reptiles originated 318–314 Ma. Our results show that reptiles emerged during the Late Carboniferous and Early Permian periods (315–292 Ma). The origin of birds is a controversial topic with several hypotheses. Some scholars believe that the earliest bird was *Protoavis* [32,33,34], whose fossils were discovered in Late Triassic strata, about 225 Ma. However, most scholars do not recognize it as a bird. Some paleontologists also support the hypothesis that birds originated from small dinosaurs. The oldest known small feathered dinosaur is *Anchiornis huxleyi* [35], whose fossils were discovered in Late Jurassic strata, about 161–151 Ma. The origin time we gave is 242–225 Ma in the Triassic period, which is consistent with the geological age of the Protoavis fossils. Our conclusion supports the hypothesis that *Protoavis* was the ancestor of birds. The origin time of today’s mammals can be traced back to the Carboniferous period, about 320–315 Ma [36,37,38]. Their ancestors were a group of animals known as non-mammalian synapsids that arose in the amniotes. The origin time of other mammals we calculated is 322–299 Ma. The molecular clock study [13] has estimated similar divergence times for major animal groups. It supports our findings for invertebrates, fishes, amphibians, reptiles and mammals.

For the Cambrian explosion event, the evidence mainly comes from animals [1], but there is very little evidence of plants. Although complex land plants did not exist during the Cambrian, we included modern plant data to verify whether the transition of genome evolution mode we proposed, that is, the phase transition from TA-independent selection to CG-independent selection, is universal in animals and plants. This approach allows us to explore whether the genomic dynamics driving the Cambrian explosion apply to a wider range of life forms, thus providing more comprehensive insights into the evolution of complex organisms. Our results showed that the evolution mechanism of animals and plants is the same. We thought that the Cambrian explosion is also reflected in the evolution of plant diversity. Here, the origin time of six plant branches were given. By comparing the origin time with paleontological evidence, we found that our results for the origin time of green algae, mosses and ferns are consistent with paleontological evidence. However, the results for the origin time of angiosperms (monocotyledons and dicotyledons) and gymnosperms differ from paleontological evidence. For green algae, the oldest fossils ever discovered are *Proterocladus antiquus* [39], which appeared 1000 Ma. The molecular clocks calibrated using these fossils suggested that green algae may have originated 1382–797 Ma during the Mesoproterozoic to Neoproterozoic or earlier [40]. The origin time of green algae we deduced is 767–712 Ma, which is close to paleontological evidence but still leaves some gaps. For mosses, paleontologists discovered the fossils of *Parafunaria sinensis Yang* dating back to 520 Ma [41]. The origin time we obtained is 456–424 Ma, which is only 64 million years later than the paleontological evidence. For ferns, paleontologists believe that the earliest fern, *Baragwanathia*, appeared between the Late Silurian and Early Devonian periods (425–395 Ma) [42]. The origin time we gave is 427–396 Ma, which is consistent with paleontological evidence.

Our results show that the origin time of angiosperms differs obviously from the paleontological evidence. The fossils of an herbaceous plant were discovered in Daohugou of China, which appeared in the Middle Jurassic period about 164 Ma [43]. However, it did not indicate whether it was monocotyledon or dicotyledon. Paleontologists discovered that *Nanjinganthus dendrostyla* gen. et sp. nov. appeared in the early Jurassic period about 174 million years ago and is considered to be the oldest flower fossil in the world [44]. Additionally, siliconized flora fossils were discovered in eastern Inner Mongolia of China, which appeared 126 Ma [45]. Furthermore, paleontologists have studied fossils of plants related to the British Jurassic and Antarctic Triassic. Their results showed that the divergence of the close groups of angiosperms (angiophytes) actually began in the late Permian (260–250 Ma), much earlier than when the crown groups of angiosperms appeared in the fossil record [45]. The fossils of a whole preserved monocotyledons plant, *Sinoherba ningchengensis*, were discovered in the Yixian Formation strata of China [46]. They appeared 125 Ma, which is the earliest reliable fossil record of monocotyledons plants in the world. The fossils of a dicotyledons plant, *Leefructus mirus*, were discovered in Lingyuan of China [47]. They appeared 124 Ma, which is the earliest dicotyledons plant fossils found so far. These paleontological fossil evidences showed that the ancestors of angiosperms may have appeared 260–250 Ma. However, the origin time we gave is 484–449 Ma for monocotyledons and 403–374 Ma for dicotyledons, which is about 200 million years earlier than the existing paleontological evidence.

Although we thought that these findings may reflect an earlier divergence of angiosperms, we acknowledged that they challenge existing fossil records. There are several reasons to support our conclusions about the origin time of angiosperms. First, most of the evidence for the Cambrian explosion appeared in animals. The fossil records of angiosperms are seriously insufficient, and the reference value is unreliable [1]. Some studies have shown that the earliest angiosperms may have appeared earlier than fossil evidence suggests [48,49]. Second, our study showed that animal and plant genome sequences follow the same evolution mechanism. Animals and plants must have a coevolution relationship in similar environments. Comparing Table 2 and Table 3, the origin time of plant branches are close to that of animal branches, and the origin time of plant branches is generally earlier than that of animal branches. Our results satisfy the coevolution rule between animals and plants. Third, considering the predator–prey relationship between animals and plants, our conclusions are more convincing. From a macro perspective, there must be a certain percentage of herbivores among various animals. Therefore, the ancestor of angiosperms should appear before the herbivores, not after them. For example, herbivorous fishes appeared about 400 Ma. What did they consume? Definitely not ferns and mosses. Critically, our analysis targets universal macro-level patterns and logical principles governing predator–prey dynamics—not species-specific interactions or micro-level mechanisms. Moreover, it is logically unacceptable that the ancestor of reptiles appeared 318–314 Ma and the ancestor of angiosperms appeared 260–250 Ma [30,31,45]. The origin time of plant branches is generally earlier than that of animal branches, indicating that our conclusions conform to the logical relationship of predator–prey. Fourth, some animals first landed ashore in 380–360 Ma [24,25,26,27,28,29] and needed food to support them. In addition to carnivores, some of the first landed animals were herbivores. Hence, plants should land before animals. After plants landed ashore, they rapidly changed the environment of the Earth’s land and created suitable conditions for the arrival of animals. Therefore, we have reason to believe that plants, probably including the ancestor of angiosperms, landed before animals. However, alternative explanations for the difference include the following: (1) limitations in our model, such as CG- and TA-independent selection intensities, which may not fully capture the evolutionary dynamics of angiosperms; (2) incomplete sampling of plant genomes, particularly for early diverging lineages; (3) potential biases in the fossil record due to preservation challenges for soft-bodied plants. These factors indicate that our deduced origin times should be viewed with caution before obtaining more genomes and paleontological evidence.

For modern gymnosperms, our results are significantly different from the paleontological evidence. Modern gymnosperms originated 158–147 Ma according to our research, while the paleontological evidence shows that gymnosperms originated 395–389 Ma [50]. We found that only the genome sequences of modern gymnosperms are in current databases, and we do not know why the other gymnosperms are not listed. Existing research results showed that gymnosperms belong to an ancient class of Spermatophyta that originated in the Devonian period of the Paleozoic era (395–389 Ma) [51,52]. Gymnosperms have mostly small needle-shaped evergreen leaves with only narrow tracheids in their vascular tissue. They grow slowly and have poor adaptability to harsh environments. Some studies suggested that most gymnosperms became extinct after several major changes in geography and climate during evolution, and only a few species survived. These species are called modern gymnosperms. They have large genome sequences, and their bodies are large, such as conifers; they can thrive in harsh environments. These factors indicate that our deduced origin time for modern gymnosperms is reasonable but may not fully represent the broader gymnosperm lineage. Further studies incorporating additional gymnosperm genomes could help clarify this difference.

The origin time of green algae was derived from the CG-independent selection intensity deduction and was found to be 767–718 Ma, which is still far from the paleontological evidence of 1382–797 Ma (Table 3). Our results showed that the TA-independent selection intensity is obviously higher in green algae, and *δ*_TA1,max_ = 6.40. For most green algae, their *δ*_TA1_ is larger than *δ*_CG1_. About 60% of green algae have *δ*_CG1_ < 1. According to the definition of *δ*_CG1_ and *δ*_TA1_, the results indicated that the evolution of green algae genome sequences is dominated by TA-independent selection mode. Therefore, it is more reasonable to select the TA-independent selection intensity to characterize the evolution state of genome sequences in green algae.

Here, we used the TA-independent selection intensity to characterize the evolution state of genome sequences in green algae. We found that the distribution of the TA-independent selection intensity is also nonlinear (Figure 3), so we used the same method to deduce the relationship between evolution state and evolution time. We assumed that the evolution rate is *v*′ = *kδ*_TA1_, and the proportional coefficient *k* is the same as that analyzed using the CG-independent selection intensity in plants. The evolution rate is defined as *v*′ = −*dδ*_TA1_/*dt*. The phase transition point is still selected as *δ*_CG1_ = 1, which corresponds to *δ*_TA1_ = 2 (Figure 3). Therefore, the starting time of the Cambrian explosion corresponding to *δ*_TA1_ = *δ*_0_′ = 2 is denoted as *t* = *t*_0_′. Finally, the evolution time *t*′ of green algae was obtained; see Equation (17). The results are shown in Figure 2C and Additional file: Appendix A. The time corresponding to the species with the highest TA-independent selection intensity (*δ*_TA1,max_ = 6.40) represents the origin time of green algae. Finally, we deduced that green algae originated 922–856 Ma, about 150 million years earlier than the origin time deduced by *δ*_CG1_. It showed that the origin time of green algae deduced by *δ*_TA1_ is more consistent with paleontological evidence [39,40]. It indicates that it is more reasonable to use the dominant evolution mode to characterize the evolution state of species genome sequences. Therefore, we must combine the two characteristic parameters of *δ*_CG1_ and *δ*_TA1_ to deduce the origin time of each plant branch.

In summary, we deduced the origin time of animal and plant branches based on our conjecture on the cause of the Cambrian explosion; our conclusion is consistent with the paleontological evidence. It verified that the phase transition process of the evolution mode of genome sequences is the cause of the Cambrian explosion.

## 4. Discussion

There are a large number of animals and plants on Earth, and our study uses a limited sample of 1087 species. Inferring ancient evolutionary events from such a sample size will definitely lead to errors. Nevertheless, our conclusions are generally consistent with paleontological evidence for animal branches and some plant branches, such as green algae, mosses, and ferns. This concordance indicates that our theoretical method captures essential aspects of genome evolution, and the proposed evolution mechanism based on CG- and TA-independent selection intensities is robust. Regarding the meta-organism model inspired by stellar evolution, we recognized that biological systems are more complex than stellar systems due to some factors such as selection pressures and extinction events. Nevertheless, the meta-organism approach provides a valuable framework for reconstructing evolutionary timelines, which is supported by its consistency with fossil records. However, further validation with additional genomes and paleontological data is needed to strengthen its theoretical foundation.

We recognize that using modern plants that emerged hundreds of millions of years after the Cambrian (such as monocotyledons and dicotyledons) may introduce time span issues. However, our research focuses on genome evolution trajectories and mechanisms, which can be used to infer the evolutionary process of their ancestors through modern species. The consistency of phase transition patterns observed in different plant branches suggests that this phenomenon may have existed in their ancient ancestors, providing indirect insights into Precambrian evolutionary dynamics. Similar methods have been widely applied in paleogenomics, such as reconstructing the evolutionary trajectories of ancient plants through the genomes of existing species [53].

In exploring the cause of the Cambrian explosion, we believed that there are still many issues that deserve further investigation. First, during the Cambrian explosion, the phase transition process in the transition region is worthy of further investigation. It is not sufficient to consider the distribution properties of 8-mer spectra of CG and TA subsets, which is reflected in the CG- and TA-independent selection intensities. It is necessary to consider the occurrence frequency of each 8-mer in CG and TA subsets and to construct a set of characteristic quantities to describe the phase transition process. Second, the evolution of genome sequences is in a peculiar state where *δ*_CG1_ > 1 and *δ*_TA1_ > 1 in the transition region; we thought that the peculiar evolution state is closely related to the species diversity that appeared during the Cambrian explosion. Therefore, the peculiar evolution state is worthy of further exploration. Some green algae appear before the phase transition region, and some green algae are in the phase transition region; mosses appear at the end or after the phase transition region. We should take green algae and mosses as a sample set to analyze the relation between the phase transition process and the Cambrian explosion. However, there are too few genome sequences of mosses in current databases. We believe that this will not be a problem in the near future.

The new evolution mechanism of genome sequences in rodents and primates is worthy of further exploration. Compared with other species, the TA-independent selection mode is weak in other mammals and disappears completely in rodents and primates. It indicated that the evolution mechanism of higher mammals has changed. Our results showed that only the 8-mer spectra of CG2, CG1 and CG0 subsets have single-peaked distribution in these genome sequences, while the other 15 classes of XY2, XY1 and XY0 subsets 8-mer spectra have triple-peaked distribution [14]. This phenomenon may be a manifestation of the new evolution mode in higher mammals. Revealing the new evolution mechanism of higher mammals is an interesting topic.

## 5. Conclusions

In our previous studies, we found the evolution mechanism of genome sequences, that is, the CG- and TA-independent selection intensities and the mutual inhibition relationship between them determine the evolution state of genome sequences. In animals and plants, the CG-independent selection intensity (*δ*_CG1_) is positively correlated, and the TA-independent selection intensity (*δ*_TA1_) is negatively correlated with the evolutionary levels of species genome sequences. Therefore, we use these two characteristic parameters to characterize the evolution state of genome sequences. From the perspective of life evolution, if animals or plants are regarded as one organism, called meta-animal or meta-plant, the sort order of species by evolutionary levels from low to high can be regarded as the evolution time of meta-animal or meta-plant. According to the evolution timeline, we found that there is a transition process from the evolution mode dominated by TA-independent selection to that dominated by CG-independent selection during the evolutionary process of lower organisms. We called this phenomenon the phase transition process, which is a revolutionary event in the evolutionary process of genome sequences. Therefore, we believed that the phase transition process is the fundamental cause of the Cambrian explosion. This conjecture assumes that the evolution mechanisms observed in modern genomes reflect those of the Cambrian period. The consistency of our results with paleontological evidence supports our conjecture. However, we acknowledge that factors such as genetic drift, lineage-specific adaptations, or incomplete sampling of modern genomes may introduce uncertainties. These uncertainties could be resolved in future studies with broader genome datasets.

Due to the nonlinear relationship between the evolution state and the evolution time, we established a theoretical model to describe the nonlinear relation. We assumed that the evolution rate is linearly positively correlated with the evolution state (*δ*_CG1_). The time corresponding to *δ*_CG1_ = 1 is defined as the starting time of the Cambrian explosion, and the time corresponding to *δ*_CG1,max_ is defined as the present time. Finally, we obtained the evolution time of animals and plants. Based on the relationship between the evolution state and the evolution time, we deduced the origin time of animal and plant branches, respectively. We found that the origin time of each animal branch is extremely consistent with paleontological evidence. In plants, the origin time of green algae, mosses and ferns is basically consistent with paleontological evidence. However, for monocotyledons and dicotyledons, the origin time is much earlier than that given by paleontology. The predator–prey and coevolution relationships between animals and plants provide the possibility of earlier plant origin. However, these findings should be interpreted cautiously due to potential model limitations, such as the reliance on CG- and TA-independent selection intensities, and gaps in genomes and fossil data. Our study provides a novel framework for exploring the Cambrian explosion through genome sequence evolution, but further studies are needed to refine estimates for certain plant lineages.

Additionally, considering the fact that the evolution of genome sequences is dominated by TA-independent selection mode in green algae, we also used the TA-independent selection intensity to deduce the origin time of green algae, and the result is closer to paleontological evidence. Our conclusion verified that the phase transition process of evolution modes of genome sequences is the cause of the Cambrian explosion, and we provide a new theoretical method to reveal the cause of the Cambrian explosion by using existing genome sequences.

## Figures and Tables

**Figure 1 biology-14-00783-f001:**
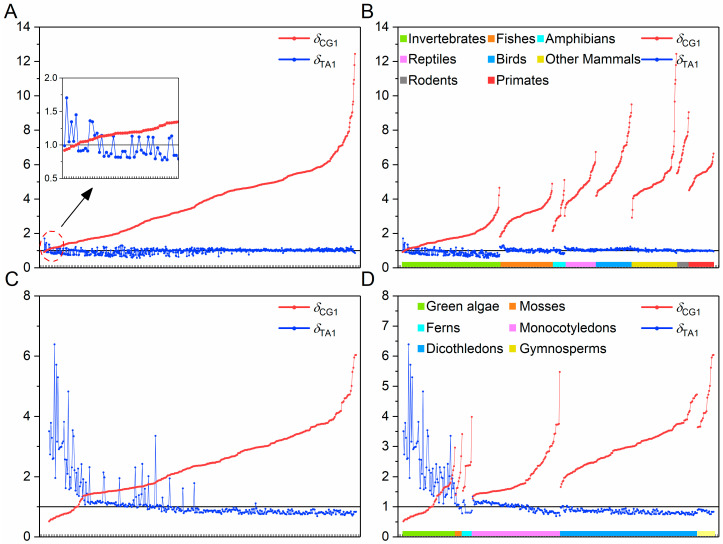
Distributions of CG-independent selection intensity (*δ*_CG1_) and TA-independent selection intensity (*δ*_TA1_) of genome sequences. The *x*-axis represents species and the *y*-axis represents the spectrum separability values. The horizontal line represents that *δ*_CG1_ = *δ*_TA1_ = 1: (**A**) Animals. Species on the abscissa are sorted in ascending order of *δ*_CG1_; (**B**) Animal branches. The branches are sorted in ascending order of evolutionary levels of animals. Within each animal branch, species are sorted in ascending order of *δ*_CG1_; (**C**) Plants. Species on the abscissa are sorted in ascending order of *δ*_CG1_; (**D**) Plant branches. The branches are sorted in ascending order of evolutionary levels of plants. Within each plant branch, species are sorted in ascending order of *δ*_CG1_.

**Figure 2 biology-14-00783-f002:**
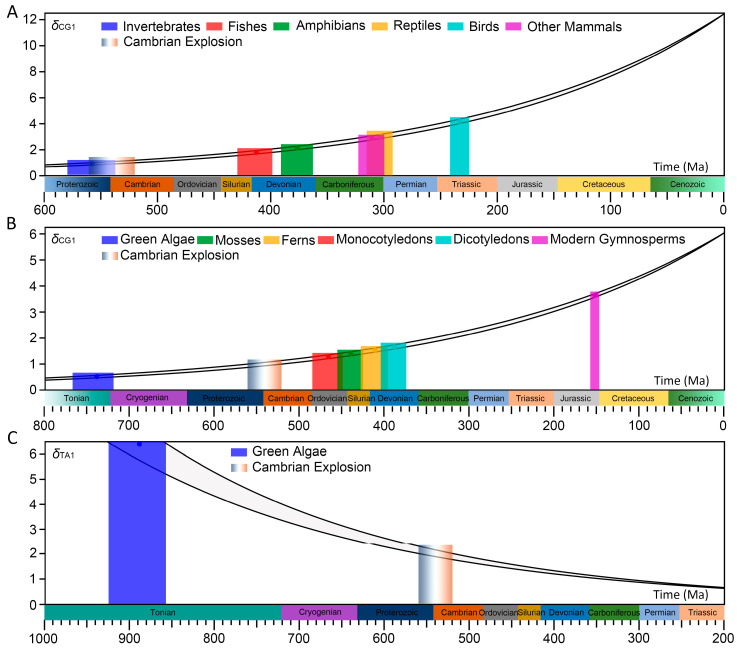
Relationship between the evolution state and the evolution time as well as the origin time of each species branch. The *x*-axis represents the evolution time and the *y*-axis represents the spectrum separability values: (**A**) Animals; (**B**) Plants; (**C**) Green algae. The relationship between the evolution state and the evolution time is characterized by TA-independent selection intensity.

**Figure 3 biology-14-00783-f003:**
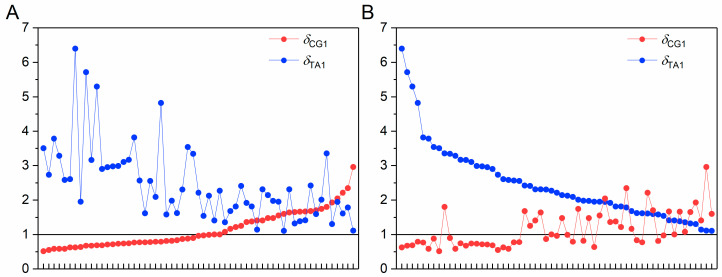
Distributions of CG- and TA-independent selection intensities in green algae. The *x*-axis represents species, and the *y*-axis represents the spectrum separability values: (**A**) Species are sorted in ascending order of the CG-independent selection intensity (*δ*_CG1_); (**B**) Species are sorted in descending order of TA-independent selection intensity (*δ*_TA1_).

**Table 1 biology-14-00783-t001:** Number of genome sequences within each branch.

Branch	Number	Branch	Number
**Invertebrates**	**230**	**Plants**	**350**
**Vertebrates**	**507**	Green algae	59
Fishes	125	Mosses	8
Amphibians	29	Ferns	11
Reptiles	74	Monocotyledons	99
Birds	85	Dicotyledons	154
Other mammals	106	Gymnosperms	19
Rodents	29		
Primates	59		

**Table 2 biology-14-00783-t002:** Comparison of origin time of animal branches with paleontological evidence (Ma).

Branch	Our Results	Paleontological Evidence	Species (Minimum *δ*_CG1_)
Invertebrates	579–537	575–542	*Dinoponera quadriceps*
Fishes	429–398	439–423	*Hippocampus comes*
Amphibians	391–363	380–360	*Rana temporaria*
Reptiles	315–292	318–314	*Sphenodon punctatus*
Birds	242–225	161–151 or 225	*Buceros rhinoceros*
Other Mammals	322–299	320–315	*Ornithorhynchus anatinus*

**Table 3 biology-14-00783-t003:** Comparison of origin time of plant branches with paleontological evidence (Ma).

Branch	Our Results	Paleontological Evidence	Species (Minimum *δ*_CG1_)
Green Algae	767–712	1382–797	*Micromonas pusilla*
Mosses	456–424	520	*Anthoceros angustus*
Ferns	427–396	425–395	*Selaginella kraussiana*
Monocotyledons	484–449	260–250	*Cenchrus americanus*
Dicotyledons	403–374	260–250	*Tarenaya hassleriana*
Modern Gymnosperms	158–147	395–389	*Larix kaempferi*

## Data Availability

The whole genome sequences and annotated information of all species involved in this study were obtained from NCBI (https://www.ncbi.nlm.nih.gov/ (accessed on 24 April 2023)) and CNGBdb (https://db.cngb.org/ (accessed on 24 April 2023)). See Additional file: Appendix A for detailed information.

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
