# Peer review of "Exploring the Causes of the Cambrian Explosion Based on the Evolution Mechanism of Genome Sequences"

_biology, 2025, doi:10.3390/biology14070783_

Round 1

Reviewer 1 Report

Comments and Suggestions for Authors

The main question addressed by this research is: What is the underlying cause of the Cambrian explosion, and can it be explained through the evolutionary mechanism of genome sequences in existing species? The authors aim to use CG- and TA-independent selection modes as a theoretical framework to infer evolutionary transitions and species origins.

Original and Contribution:

The topic is original and relevant within the fields of evolutionary biology and genomics. The study attempts to fill a gap by proposing a genome sequence-based mechanism as the intrinsic driver of the Cambrian explosion, complementing existing theories that emphasize environmental or ecological factors. This genomic perspective provides a novel methodological approach and theoretical insight. However, the paper would benefit from a clearer positioning of its novelty compared to recent molecular clock studies and genomic analyses.

This study adds a new theoretical model linking genome sequence evolution modes with major evolutionary events, proposing a phase transition from TA to CG-independent selection dominance as a fundamental cause of the Cambrian explosion. This hypothesis extends previous work on genome evolution and offers a quantitative framework to estimate species divergence times based on sequence features. Compared with other published materials, it provides an integrative perspective bridging molecular evolution and paleontological timelines.

Methodological improvements 

  1. The choice of genome samples (species coverage and selection criteria) should be better justified, especially given the uneven availability of genome data across taxonomic groups.
  2. Statistical validation or robustness analysis of the separability and conservatism parameters would strengthen confidence in the inferred selection intensities.
  3. The assumption of linearity or proportionality in evolutionary rate models should be discussed or tested against alternative models.
  4. Integration of additional molecular markers might enhance the phylogenetic inferences.

Consistency of conclusions with evidence :

The conclusions are generally consistent with the evidence and arguments presented, although some claims  would benefit from more cautious interpretation. While the model aligns well with paleontological data for several animal and plant groups, discrepancies—especially for angiosperms and gymnosperms—should be discussed in more detail, including alternative explanations or data limitations.

The references are mostly appropriate and cover both classical and recent literature on the Cambrian explosion and molecular evolution. A few additional references from recent high-impact journals on molecular clocks, phylogenomics, and genome evolution mechanisms could be added to strengthen the background and comparison sections.

Figures are clear and informative; however, some figure legends could provide more explanation of axes and parameter definitions for readers less familiar with 8-mer spectral analysis. 

Reviewer 2 Report

Comments and Suggestions for Authors

114- I was unfamiliar with the term "life genome sequences". While the statement is correct, there may be other readers also familiar with the term. Your choice to modify or leave as is, simply sharing a thought. 

Below are three grammar suggestions

343- conservatism may not be the best word choice. 

548-gymnosperms "are not listed". 

563- About 60% of green algae have a CG1 <1. 

Methods, math, and assumptions are clearly stated and justified. 

The divergence of green algae from the plant and animal trends were apparent from the first figures and the higher GC content displayed in the supplemental data. Due to these observations, and other algae-specific deviations mentioned in the manuscript, I think Figure 3 was necessary to address the difference. 

While this manuscript was informative and pioneering, it left me wondering why green algae showed a unique trend. Furthermore, what factors drove primates and rodents to adopt a new evolution mechanism. Are there any genetic insights as to what mechanism was dominate for algae or for primates/rodents? These hypothesis may be outside the scope of this manuscript maybe these ideas could be addressed as a trilogy with this manuscript and the publication identifying GC- and AT-independent selection. Could CG- or AT-dependence account for the divergence? 

Overall well written manuscript. 

Comments on the Quality of English Language

Overall quality of English was sufficient. A few grammatical suggestions were provided. 

Reviewer 3 Report

Comments and Suggestions for Authors

This manuscript presents an ambitious and creative attempt to investigate the cause of the Cambrian explosion through patterns of genome sequence evolution in extant species. The authors analyze dinucleotide-based k-mer frequency distributions (specifically CG- and TA-independent selection modes) across over 1000 genomes and propose that a transition from TA to CG dominance may have catalyzed the explosion of biodiversity in the Cambrian period. This is an intriguing hypothesis that connects molecular evolution with macroevolutionary events.

However, the manuscript in its current form has several substantial issues—both conceptual and technical—that must be addressed before it can be considered for publication. 

First, I must say that I am not a specialist in k-mer frequency analysis or genomic spectral modeling. Therefore, I cannot fully assess the statistical robustness of the selection intensity metrics used (δCG1 and δTA1).

The most important thing to check in the draft and rewrite, from my perspective, is the inclusion of plant data—especially flowering plants and gymnosperms—in the analysis of Cambrian evolution. Since complex land plants did not yet exist during the Cambrian period, their role in modeling genome evolution relevant to this event is questionable. I understand that the authors aim to build a generalized model of genomic evolution using extant species as a proxy for an evolutionary trajectory, but this cross-domain assumption (i.e., treating plant and animal evolution as governed by identical genomic selection dynamics) is not sufficiently supported.Furthermore, using plant taxa that appeared hundreds of millions of years after the Cambrian (e.g., monocotyledons, dicotyledons) to reconstruct pre-Cambrian or Cambrian conditions may introduce significant temporal confounds. The authors should justify this inclusion and clearly explain its limitations.

L522-543 in the draft (The entire text lacks references to any literature and relies solely on the presentation of its own viewpoints without providing objective evidence for support.), The manuscript hypothesizes that angiosperms originated earlier than key animal groups based on four arguments, but the reasoning contains fundamental conceptual and chronological errors requiring revision. The central flaw lies in conflating angiosperms (flowering plants, ~140–130 Ma) with early land plants (e.g., bryophytes, ~475 Ma;  ferns, ~425 Ma), leading to invalid ecological and evolutionary claims.  For instance, the assertion that angiosperms arose 260–250 Ma directly conflicts with Cretaceous fossil evidence (e.g., Archaefructus) and molecular data, as this timeframe corresponds to gymnosperm dominance.  Similarly, linking herbivorous animals (e.g., Devonian placoderms) to angiosperms is erroneous—these animals fed on algae, bryophytes, and early vascular plants, not angiosperms, which appeared 200 million years later.  The text further misinterprets terrestrial colonization timelines: while non-vascular plants indeed preceded animal land colonization, angiosperms themselves emerged long after tetrapods.  Methodological issues include unsubstantiated comparisons of plant-animal genomic evolution (ignoring critical differences like whole-genome duplications in plants) and citation inaccuracies (e.g., Permian reptiles were carnivorous, and Triassic flora was gymnosperm-dominated).  To salvage the hypothesis, authors must rigorously distinguish angiosperms from earlier plant groups, align arguments with Cretaceous angiosperm fossil records, and address ecological realities (e.g., Carboniferous ecosystems thrived on ferns/lycophytes without angiosperms).  A restructuring focusing on validated coevolutionary mechanisms (e.g., Cretaceous pollinator interactions) and updated citations is essential. Without correcting these errors, the manuscript’s core premise remains untenable.

Besides, some other concerns:

  1. The biological mechanism connecting k-mer frequency patterns to the morphological and ecological complexity of the Cambrian explosion is underexplained. How shifts in dinucleotide representation drive multicellularity or body plan innovation remains speculative. Extrapolating from current genomes to deep evolutionary timelines risks circular reasoning.
  2. The assumption that modern genome sequences can reliably infer ancient states requires explicit justification with appropriate caveats. The analogy to stellar evolution (meta-organism modeling) lacks a strong theoretical foundation in evolutionary biology. It appears overextended without sufficient biological precedent.
  3. There are several instances of uncritical or inaccurate citations. For example, the manuscript references ancient fossil taxa without consistently citing primary paleontological literatures. (eg: L57-59, L69-71, L85-86, L93-94 etc.)
  4. some genus and species names throughout the manuscript (e.g., Funisia Dorothea) are not italicized as per standard biological formatting conventions.
Comments on the Quality of English Language

NONE

Round 2

Reviewer 3 Report

Comments and Suggestions for Authors

This manuscript presents a novel and ambitious hypothesis linking the Cambrian explosion to a genomic transition from TA-biased to CG-biased selection mechanisms. While the authors have responded to my previous comments and made textual revisions, several critical conceptual and methodological flaws remain unaddressed, which significantly undermine the scientific validity of the study. Unless these major issues are directly and substantially addressed, the manuscript is not suitable for publication in its current form.

Below, I outline the most serious issues:

  1. Temporal Misalignment Between Research Objective and Taxon Sampling

The central aim of this manuscript is to explore potential genomic mechanisms underlying the Cambrian explosion. However, the taxonomic sampling includes numerous post-Cambrian taxa—such as birds, reptiles, and flowering plants—which did not exist during the Cambrian period. Their inclusion introduces severe temporal and evolutionary inconsistencies that compromise the integrity of the analysis.

The Cambrian explosion was characterized by the rapid diversification of invertebrate metazoan phyla, including arthropods, annelids, mollusks, and cnidarians. Yet these groups receive minimal attention or are aggregated into a single “invertebrate” category, while more analytical emphasis is given to vertebrates that evolved hundreds of millions of years later. Similarly, the inclusion of modern plant taxa, especially angiosperms that diversified long after the Cambrian (~130 Ma), introduces further temporal incongruence.

This represents a core conceptual flaw. If the goal is to understand genome evolution relevant to Cambrian biodiversity, the analysis must focus on early-diverging animal phyla. The inclusion of Mesozoic and Cenozoic taxa without strong justification fundamentally undermines the study’s relevance.

I strongly recommend restructuring the dataset to highlight early-branching invertebrate phyla. These groups should be categorized separately (e.g., arthropods, annelids, and cnidarians), and later-appearing vertebrate and plant taxa should either be excluded or explicitly justified for their inclusion.

  1. Misrepresentation of Angiosperm Origins and Fossil Evidence

The manuscript continues to suggest an angiosperm origin around 260–250 Ma, which contradicts robust paleobotanical evidence placing the earliest known flowering plants in the Early Cretaceous (~130–125 Ma), with fossils such as Archaefructus. Furthermore, the speculative link between early herbivorous animals and angiosperms remains ecologically and temporally implausible.

Despite the authors' claim that this section is "exploratory," the current framing still implies causality that is not supported by fossil or molecular data.

This section must be thoroughly revised. The authors must clearly distinguish between angiosperms and early land plants (e.g., bryophytes, lycophytes) and remove ecological arguments that conflict with established paleobotanical timelines.

  1. Lack of Mechanistic Explanation Linking k-mer Biases to Evolutionary Innovation

While the authors assert that CG/TA dinucleotide frequency shifts are correlated with species evolution, the manuscript still lacks a biologically grounded mechanism explaining how such shifts contribute to key Cambrian traits such as body plan complexity, developmental modularity, or regulatory innovations.

Lines 383–406 refer to “special evolutionary states” (e.g., δCG1 > 1, δTA1 > 1), but their biological significance is vague. It remains unclear whether these genomic signatures correlate with known events such as Hox gene duplication, gene family expansions, or the emergence of novel regulatory elements.

Currently, the model remains correlative and descriptive, rather than explanatory or predictive.

  1. Conceptual and Ecological Flaws in the Use of Modern Plant Genomes

The authors now acknowledge that angiosperms (flowering plants) originated long after the Cambrian period. However, they continue to justify the inclusion of angiosperm genomes in their analysis using deeply flawed ecological reasoning—specifically, that early herbivorous animals required angiosperms as a food source (Lines 592–599). This claim is both temporally and ecologically unsupported.

Early terrestrial herbivores evolved well before the emergence of flowering plants, feeding primarily on algae, bryophytes, lycophytes, and early ferns. The authors conflate "land plant colonization" with "angiosperm origins" , collapsing a ~300-million-year gap and misrepresenting plant-animal co-evolutionary dynamics.

More seriously, the argumentation suffers from circular reasoning. The manuscript uses its own disputed molecular dates—suggesting that plant lineages originated before animals—as the basis for asserting a general ecological “rule” of plant-first emergence. This “rule” is then used to validate the same molecular dates, creating a self-referential loop that lacks independent empirical grounding from functional genomics, paleobotany, or fossil records.

Additionally, the authors cite Devonian herbivorous fish as supporting evidence for the early appearance of angiosperms. This is biologically untenable—these fish fed on non-vascular algae or primitive vascular plants, not on angiosperms, which did not exist until the mid-Mesozoic. Similarly, Permian herbivores like dicynodonts thrived on gymnosperms, not flowering plants. The authors’ predator–prey logic lacks evidentiary support and ignores well-established Mesozoic and Paleozoic ecosystem structures.

Terminological imprecision compounds the problem. The term “coevolution” is misused to describe mere temporal sequence, without evidence of reciprocal adaptation between lineages—a defining criterion of coevolutionary theory.
